# Rapid and sensitive detection of SARS-CoV-2 infection using quantitative peptide enrichment LC-MS analysis

Andreas Hober[1], Khue Hua Tran-Minh[1,2], Dominic Foley[3], Thomas McDonald[3], Johannes PC Vissers[3], Rebecca Pattison[3], Samantha Ferries[3], Sigurd Hermansson[3], Ingvar Betner[3], Mathias Uhlén[1,2], Morteza Razavi[4], Richard Yip[4], Matthew E Pope[4], Terry W Pearson[4], Leigh N Andersson[4], Amy Bartlett[3], Lisa Calton[3], Jessica J Alm[5], Lars Engstrand[6], Fredrik Edfors[1,2]*

[1]Science for Life Laboratory, Solna, Sweden; [2]The Royal Institute of Technology, Division of Systems Biology, Department of Protein Science, School of Chemistry, Biotechnology and Health, Stockholm, Sweden; [3]Waters Corporation, Milford, United Kingdom; [4]SISCAPA Assay Technologies, Inc, Victoria, Canada; [5]Karolinska Institutet, Department of Microbiology, Tumor and Cell Biology & National Pandemic Center, Karolinska Institutet, Solna, Sweden; [6]Microbiology, Tumour and Cell Biology, Karolinska Institutet, Stockholm, Sweden

*For correspondence: fredrik.edfors@scilifelab.se

**Abstract** Reliable, robust, large-scale molecular testing for severe acute respiratory syndrome coronavirus 2 (SARS-CoV-2) is essential for monitoring the ongoing coronavirus disease 2019 (COVID-19) pandemic. We have developed a scalable analytical approach to detect viral proteins based on peptide immuno-affinity enrichment combined with liquid chromatography-mass spectrometry (LC-MS). This is a multiplexed strategy, based on targeted proteomics analysis and read-out by LC-MS, capable of precisely quantifying and confirming the presence of SARS-CoV-2 in phosphate-buffered saline (PBS) swab media from combined throat/nasopharynx/saliva samples. The results reveal that the levels of SARS-CoV-2 measured by LC-MS correlate well with their correspondingreal-time polymerase chain reaction (RT-PCR) read-out (r = 0.79). The analytical workflow shows similar turnaround times as regular RT-PCR instrumentation with a quantitative read-out of viral proteins corresponding to cycle thresholds (Ct) equivalents ranging from 21 to 34. Using RT-PCR as a reference, we demonstrate that the LC-MS-based method has 100% negative percent agreement (estimated specificity) and 95% positive percent agreement (estimated sensitivity) when analyzing clinical samples collected from asymptomatic individuals with a Ct within the limit of detection of the mass spectrometer (Ct ≤ 30). These results suggest that a scalable analytical method based on LC-MS has a place in future pandemic preparedness centers to complement current virus detection technologies.

## Editor's evaluation

With the prospect of pandemic readiness, the data presented here shows that MS can and most probably will start to become an essential analytical contribution.

## Introduction

The severe acute respiratory syndrome coronavirus 2 (SARS-CoV-2) (*Wu et al., 2020*), leading to the coronavirus disease 2019 (COVID-19), has had a significant impact on human health globally, with

more than 234 million confirmed cases (*Dong et al., 2020*), assessed October 4, 2021. The effects of the pandemic are devastating and have led to lockdowns of urban areas across the globe as a response to contain any potential outbreaks (*Hale et al., 2021*). To monitor the disease, huge investments have been directed toward infrastructure for large-scale testing for ongoing COVID-19 infection (*Baker et al., 2020*). Population-wide screening or cohort testing in the vicinity of an outbreak epicenter is an essential pillar in the global fight against COVID-19 and an indispensable contribution to currently ongoing vaccination programs that pave the way for re-opening societies when entering the endemic phase. Thus, specific molecular diagnostic tools suitable for efficient disease monitoring will play a key role when countries slowly lift their bans on public gatherings, events, and global travel.

The diagnostic method called real-time polymerase chain reaction (RT-PCR) (*Freeman et al., 1999*) is the most widely used technology for detecting SARS-CoV-2 and was established within days after the virus genome was released (*Corman et al., 2020*). The method is considered as the gold standard by WHO for diagnosing patients with COVID-19 in routine clinical practice. Large-scale laboratories dedicated to PCR-based diagnostics rapidly mobilized worldwide in the early phase of the pandemic, which led to a sudden global shortage of diagnostic reagents (*Woolston, 2021*). The PCR tests generally have high analytical sensitivity and specificity, even for self-collected samples, often in the range of 95–100% (*Altamirano et al., 2020*) when evaluated in clinical settings. The observed variance between tests can be partly explained by the inherent sensitivity of the PCR reaction itself or by pre-analytical biases (*Lippi et al., 2020*), which could lead to either false-positive or false-negative results. For example, the viral genes can be amplified to detect the virus within days of infection, but the high sensitivity has also been subjected to criticism since it can detect genetic material in circulation not only days after but also multiple weeks after the first day of symptom onset (*Lan et al., 2020*). The current level of the clinical false-positive rate associated with PCR tests is unknown but is dependent on what type of PCR kit and criteria have been used. Some studies report that it can be as much as 4% at certain test facilities (*Surkova et al., 2020*). This type of error has the potential to cause the most harm in a scenario entering post COVID-19 when large-volume screening is performed in communities with low prevalence (*Healy et al., 2021*).

As a response to the global shortage, rapid antigen tests have been deployed that directly detect viral antigens. These rapid tests show similar specificity to PCR-based assays (*Weissleder et al., 2020*), but several studies have shown that they lack sufficient sensitivity when compared to RT-PCR (*Fitzpatrick et al., 2021*; *Perchetti et al., 2021*). Antigen tests also require affinity reagents, an initial bottleneck and a significant hurdle to overcome in the initial phase of a pandemic, but can scale massively once they have been generated. Additionally, rapid tests only provide a binary read-out (positive/negative), which can be difficult to interpret and the antigen is rarely specified (*Sethuraman et al., 2020*). Due to their rapid turnaround and affordability, these tests can thus be deployed in millions to aid in large-scale screening efforts and by repeated testing over time, accuracy can be greatly improved (*Mina et al., 2020*; *Ramdas et al., 2020*).

In contrast to traditional PCR tests or antigen rapid tests, LC coupled to multiple reaction monitoring (MRM) tandem MS detection offers a straightforward assay toward pre-defined targets. Turning to MS measurements to detect SARS-CoV-2 in samples directly addresses the issue of specificity and the risk of returning false-positive results as the measurement benefits from the fundamental properties of MS detection of peptides through multiple specific product ions (*Gillette and Carr, 2013*), which results in sequence-based specificity through direct physical detection of analyte molecules. The instrumentation provides reliable quantification for absolute protein concentration determination and modern MS instrumentation offers unsurpassed specificity, high precision, excellent quantitative performance, and high analytical sensitivity.

When combining these features with affinity reagents, such as antibodies, assays can reach very high sensitivities and low levels of a protein can be detected even in complex matrices. The combination of immuno-based strategies with MS read-out can complement each other and provide target-specific protein quantification (*Whiteaker and Paulovich, 2011*). In fact, it is an ideal combination for rapid detection and reliable quantification of low abundance proteins. Stable isotope labeled (SIL) standards and capture by anti-peptide antibodies (SISCAPA) (*Anderson et al., 2004*) enables multiplexed analysis of pre-digested clinical samples using peptide-reactive antibodies, selective for SARS-CoV-2 peptides, immobilized onto magnetic beads. Additionally, spiked SIL peptide standards further improve precise protein measurements performed by MRM (*Brun et al., 2009*). The use of LC-MS for

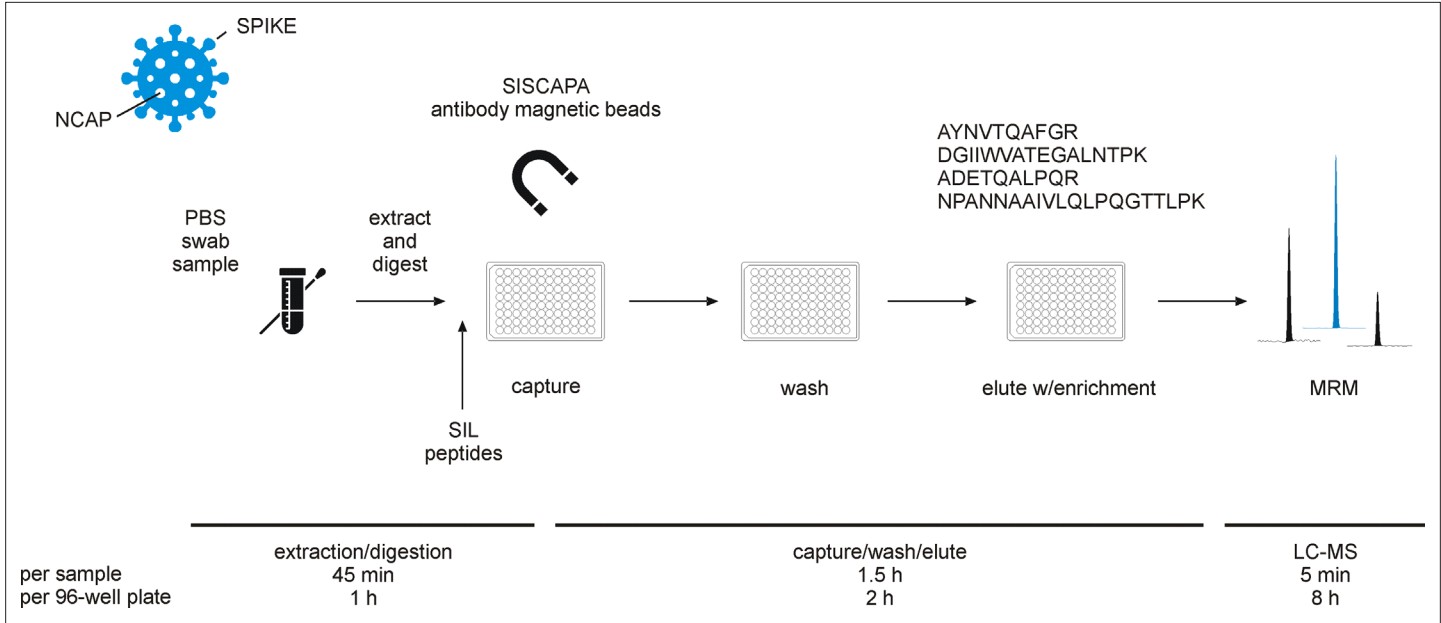

**Figure 1.** Experimental workflow for immuno-affinity peptide (stable isotope labeled [SIL] standards and capture by anti-peptide antibodies [SISCAPA]) enrichment liquid chromatography-mass spectrometry (LC-MS) of nucleocapsid protein (NCAP) severe acute respiratory syndrome coronavirus 2 (SARS-CoV-2) peptides. Swab sample extracts were subjected to tryptic digestion, SIL standards added to the tryptic digest solution, and magnetic beads coupled with specific anti-peptide antibodies incubated to allow binding of the peptides. Unbound peptides are removed and the target peptides eluted and measured using multiple reaction monitoring (MRM) analysis with LC-MS.

protein quantification of SARS-CoV-2 peptides eliminates the dependence on PCR reactions and any issues related to unspecific amplification, thanks to the selectivity achieved at three different levels: first by the antibody; second by the mass spectrometric read-out, and finally, the internal standard. As a proof of concept, we analyzed clinical samples collected from asymptomatic individuals screened for ongoing disease by RT-PCR. Samples were taken from the upper respiratory tract (combined triple-point collection strategy throat/nasopharynx/saliva) and a set of 48 PCR positive and 308 RT-PCR negative samples were selected for LC-MS analysis. All samples were analyzed using the SISCAPA immuno-affinity peptide enrichment protocol followed by LC-MS read-out outlined in *Figure 1*. The application of immuno-affinity peptide enrichment is typically associated with the detection of protein disease markers in body fluids, such as plasma or dried blood spot samples. Here, the novel application of the technology is demonstrated to detect and quantify infection by analyzing the protein complement of viruses at relevant levels, which are proven difficult to reach without enrichment (*Van Puyvelde et al., 2021*). This study thereby presents a precise and complementary approach to RT-PCR to reliably detect SARS-CoV-2 in a research or clinical setting and a possible route forward to support population-wide screening.

## Results

The application of LC-MS to detect tryptic digest peptides of SARS-CoV-2 proteins has been successfully demonstrated (*Cardozo et al., 2020*; *Cazares et al., 2020*; *Freire-Paspuel and Garcia-Bereguiain, 2021*; *Gouveia et al., 2020a*; *Gouveia et al., 2020b*; *Ihling et al., 2020*; *Saadi et al., 2021*; *Van Puyvelde et al., 2021*). However, these studies also highlight that the technique can be hampered by matrix effects, that is, analysis interferences arising from the constituent components of swab (preservation) media or other matrices, as well as base sensitivity, to be able to reach clinically relevant detection levels, suggesting the need for clean-up, for example, solid phase-based extraction and/or affinity enrichment (*Renuse et al., 2020*; *Van Puyvelde et al., 2021*). Moreover, commonality can be observed within the results of these studies in terms of which tryptic digest peptides are typically detected by means of LC-MS. Nucleocapsid protein (NCAP) is the most abundant viral SARS-CoV-2 protein with an estimated ~300–1000 copies per virion particle (*Bezstarosti et al., 2020*;

*Phimister et al., 2020*), making it, because of the relatively high number of NCAP copies per virion, an attractive target for LC-MS-based detection compared to other viral proteins. A number of NCAP candidate peptides were therefore evaluated in terms of enrichment efficiency and LC-MS behavior, that is, sensitivity and linear dynamic range, and peptide immunoassay suitability (*Whiteaker et al., 2011*). The LC-MS MRM responses of a number of candidate NCAP SIL peptides are shown in *Figure 2—figure supplement 1*, ranking the peptides in descending order of MRM sensitivity. From this set of peptides, primarily based on both MRM response and peptide immunoassay suitability, peptide AYNVTQAFGR was found to be one of the best surrogate peptide candidates, but, equally importantly, it is not significantly affected to date by known SARS-CoV-2 virus mutations (https://www.gisaid.org/). Other evaluated peptides, but not discussed in detail, included ADETQALPQR, DGIIWVATEGALNTPK, and NPANNAAIVLQLPQGTTLPK, of which the basic quantitative characterization results are summarized in *Figure 2—figure supplements 2–4*, respectively.

## Method characterization

The LC-MS MRM data were processed using TargetLynx XS and with a cut-off threshold algorithm based on peptide peak height and area thresholds, as well as quantifier to qualifier ion ratio threshold (30%). In other words, using two different consistently measured peptide fragment ions, that is, MRM transitions, to confirm the presence of SARS-CoV-2 proteins. Typical detection examples for the quantifier, qualifier, and SIL MRM transitions are shown in the (A) panel of *Figure 2*. An internal standard SIL corrected LC-MS calibration curve for antibody enriched NCAP peptide AYNVTQAFGR detected in a spiked nasopharyngeal swab matrix solution is shown in the (B) panel of *Figure 2*, covering a linear dynamic range from 3 to 50,000 amol/µl, providing >4 orders of linear dynamic range, meanwhile affording an LLOQ amount of 3 amol/µl of AYNVTQAFGR peptide (with precision ≤20%, bias ±20 % and S/N > 10:1 [peak-to-peak]). Shown as well are example quantifier and qualifier MRM chromatograms of positive (*Figure 2C*) and negative (*Figure 2D*) SARS-CoV-2 phosphate-buffered saline (PBS) swab samples. The selectivity of the method is highlighted by the complete absence of signal in the MRM chromatogram of the negative SARS-CoV-2 sample (*Figure 2D*).

The precision of the method was evaluated at 3, 10, 400, and 25,000 amol/µl for NCAP AYNVTQAFGR peptide and NCAP spiked into PBS and viral transport medium (VTM, Liofilchem, Italy). Peptides were enriched by antibodies and samples were analyzed in replicates of 5-over-5 separate occasions. The inter- and intra-day precision values of the method, as summarized in *Table 1*, were shown to be ≤20 % CV.

Additionally, the AYNVTQAFGR peptide was shown to be stable in the autosampler at 10°C for over 48 hr following re-analysis and comparison to a stored calibration curve.

## Sample analysis

The samples analyzed by LC-MS and RT-PCR were compared. The high and low pools were analyzed in triplicate with a precision of 3.0% CV and 12.2% CV, respectively, for each pool. Example quantifier and qualifier LC-MS MRM chromatograms of peptide AYNVTQAFGR are shown in the two bottom panes of *Figure 2*, respectively. The results shown in *Figure 3A* suggests good (inverse) correlation between the LC-MS ($\log_2$ transformed quantifier response, i.e., SIL corrected quantifier peak area) and the RT-PCR (Ct) data, which has also been noted in other so-called 'non-enriched' studies (*Van Puyvelde et al., 2021*). The results shown in *Figure 3B* represent the LC-MS data in an alternative, quartile distribution-based format, suggesting that differentiation between sample types is feasible and that the detected abundances are significantly different (p = 0.00018; Mann-Whitney U test).

Following CLSI EP 12-A2 User Protocol for Evaluation of Qualitative Test Performance guidance, a summary of the sample analysis results is shown in a 2 × 2 contingency table format in *Figure 4*, using the RT-PCR results as a reference, estimated sensitivity and specificity values for LC-MS are 83.3% and 100%, respectively. The 95% score confidence interval (CI) limits for sensitivity calculations were 70.4–91.3% and for specificity were 98.8–100%. Accordingly, the agreement between RT-PCR and LC-MS was strong (kappa value of 0.9 [95% CI 0.83–0.97]). When analyzing samples above the estimated LLOQ (3 amol/µl, which approximates to Ct ≤30), the estimated sensitivity is improved to 94.7% with the corresponding 95% score CI limits for sensitivity 82.7–98.5%. Further work will look at adding a secondary confirmatory peptide to the cut-off algorithm. However, RT-PCR does not distinguish between infectious virus and non-infectious nucleic acids (*Engelmann et al., 2021*),

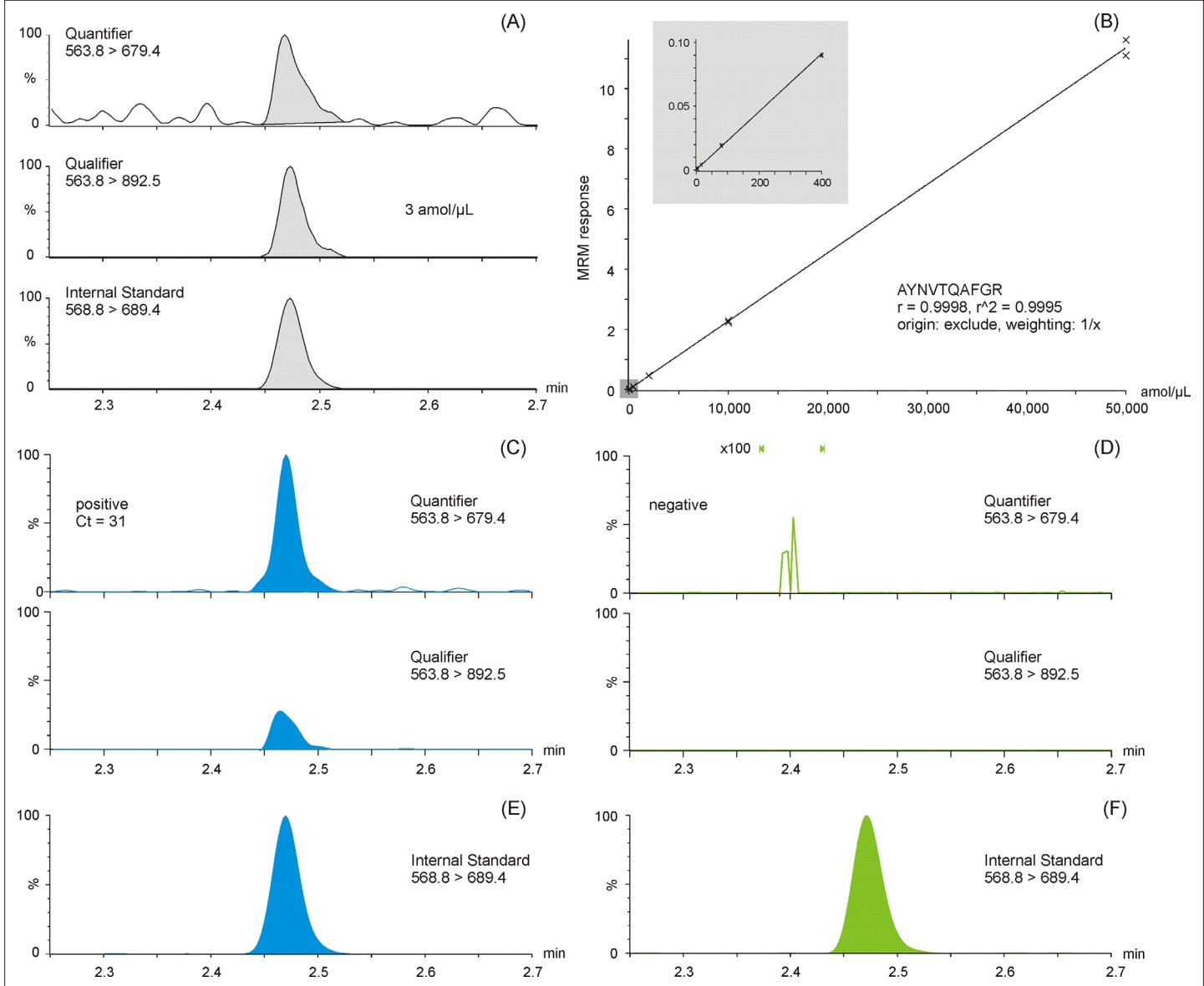

**Figure 2.** Multiple reaction monitoring (MRM) chromatograms of antibody enriched nucleocapsid protein (NCAP) AYNVTQAFGR peptide. Quantifier, qualifier, and stable isotope labeled (SIL) internal standard peptide chromatograms spiked at the lower limit of quantification (3 amol/µl) (**A**). Calibration curve of the AYNVTQAFGR peptide based on enriched recombinant NCAP digest, spiked with a constant amount of SIL peptide (**B**). Two representative intensity-scaled MRM chromatograms of positive (mean cycle threshold [Ct] 31) (**C**) and negative (blank) (**D**) severe acute respiratory syndrome coronavirus 2 (SARS-CoV-2) swab samples, respectively, normalized to the most abundant shared MRM transition. Intensity-scaled SIL internal standard peptide MRM chromatograms of positive (**E**) and negative (**F**) SARS-CoV-2 swab samples.

The online version of this article includes the following figure supplement(s) for figure 2:

**Figure supplement 1.** Peak area (multiple reaction monitoring [MRM] sensitivity) of stable isotope labeled (SIL) ($^{13}C_6$$^{15}N_2$ C-terminal K or $^{13}C_6$$^{15}N_4$ C-terminal R labeled) nucleocapsid protein (NCAP) peptides as function of peptide and detergent (CHAPS) concentration.

**Figure supplement 2.** Calibration curve for ADETQALPQR over the range 3–50,000 amol/µl.

**Figure supplement 3.** Calibration curve for NPANNAAIVLQLPQGTTLPK over the range 3–50,000 amol/µl.

**Figure supplement 4.** Calibration curve for DGIIWVATEGALNTPK over the range 3–2000 amol/µl.

whereas LC-MS will only detect one or multiple peptides from the protein complement of the virus. This has implications on the interpretation of RT-PCR Ct levels itself in terms of infectious vs. non-infectious classification of patient samples but also for determining the sensitivity and specificity of complementary and/or alternative methods. Peptide levels have not been evaluated in the context of

**Table 1.** Intra- and inter-day method precision (n = 5) when monitoring severe acute respiratory syndrome coronavirus 2 (SARS-CoV-2) nucleocapsid protein (NCAP) peptide AYNVTQAFGR using immuno-affinity peptide enrichment liquid chromatography-mass spectrometry (LC-MS) (multiple reaction monitoring [MRM]).

| | Precision (% CV) | | | | | | | |
| --- | --- | --- | --- | --- | --- | --- | --- | --- |
| | Intra (concentration [amol/µl]) | | | | Inter (concentration [amol/lµl]) | | | |
| | 3 | 10 | 400 | 25,000 | 3 | 10 | 400 | 25,000 |
| Peptide-spiked PBS | 12.0 | 11.1 | 5.8 | 5.2 | – | – | – | – |
| NCAP-spiked PBS | 18.9 | 3.9 | 4.8 | 6.4 | – | – | – | – |
| Peptide-spiked VTM | 12.5 | 6.8 | 2.4 | 3.0 | 15.5 | 10.2 | 6.8 | 4.7 |
| NCAP-spiked VTM | 13.2 | 10.2 | 2.4 | 2.9 | 11.6 | 17.6 | 18.5 | 11.1 |

–, not tested.

infectiousness yet, but other conditions, such as sample storage before LC-MRM/MS, can also give rise to analytical variance due to the inherent difference in stability between RNA and proteins. Additionally, Ct values are not universally applicable as they differ between manufacturers and methods (*Engelmann et al., 2021*; *van Kasteren et al., 2020*), which enforces the need of methods that are capable of determining viral load more accurately.

## Discussion

Any diagnostic test result should be interpreted in the context of the probability of disease, but also include proper internal controls to ensure a high level of clinical specificity when used as a tool for large-scale screening. In this conceptual study, we have established an assay capable of detecting SARS-CoV-2 in self-collected samples. Multiple peptide assays were generated toward the NCAP and the assay that gave the best response was used to profile SARS-CoV-2 in clinical samples. The unsurpassed specificity of mass spectrometers combined with antibodies is an attractive route forward for a future molecular pandemic surveillance system. This specificity can help grow the assay repertoire, which can be expanded to cover multiple peptides or proteins by rather simple means, if anti-peptide antibodies are available. The SISCAPA peptide enrichment method ensures both high sensitivity and low risk of reporting false positives due to the combination of specific binding of the antibody

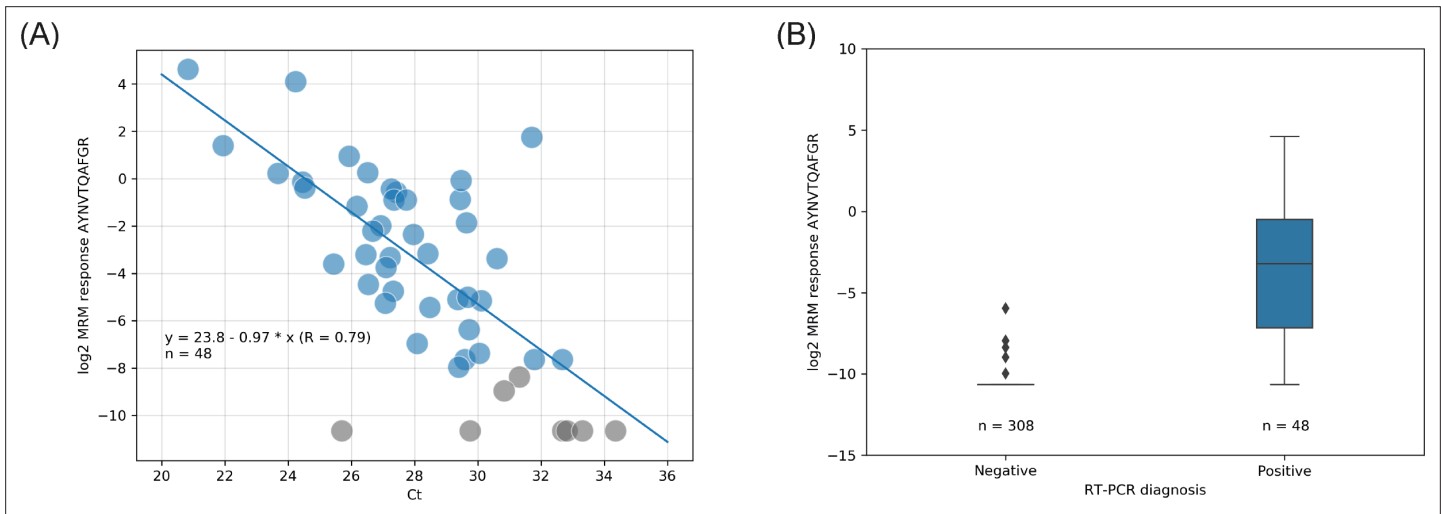

**Figure 3.** Liquid chromatography-mass spectrometry (LC-MS) (log$_2$ quantifier response) vs. real-time polymerase chain reaction (RT-PCR) (cycle threshold [Ct]) read-out correlation with linear regression (**A**) and quartiles distribution of the LC-MS results (**B**). Color labeling is based on RT-PCR diagnoses; blue = positive severe acute respiratory syndrome coronavirus 2 (SARS-CoV-2); gray = not detected (no light signals) or inconclusively quantified (single transition) by LC-MS.

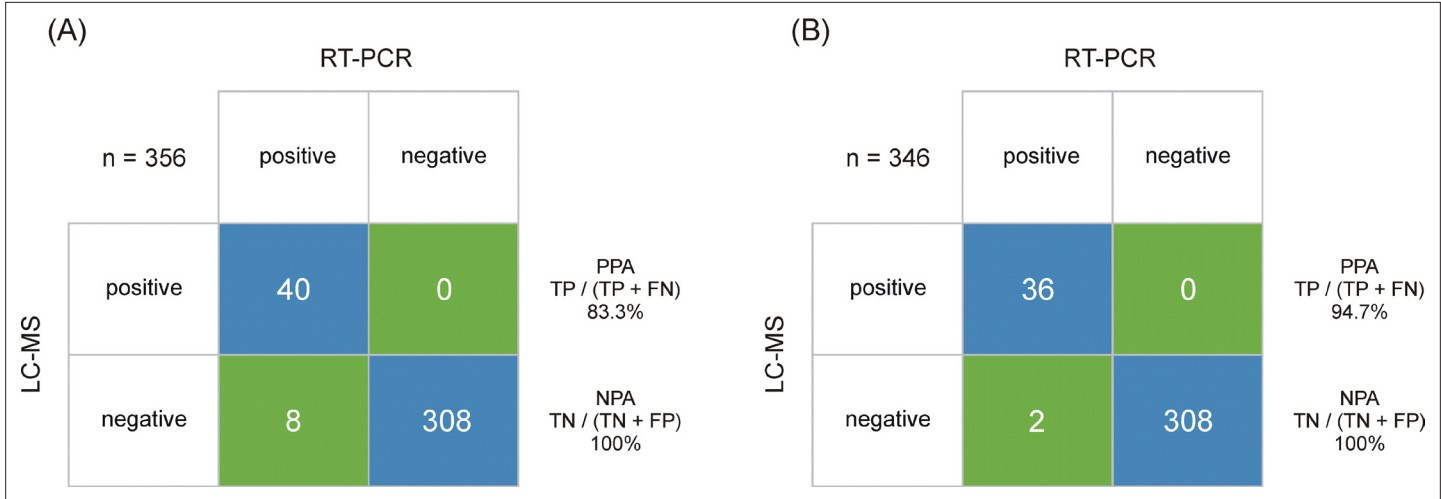

**Figure 4.** Output class (liquid chromatography-mass spectrometry [LC-MS]) vs. target class (real-time polymerase chain reaction [RT-PCR]) contingency matrix, used to calculate the positive percent agreement (PPA) and negative percent agreement (NPA) of the severe acute respiratory syndrome coronavirus 2 (SARS-CoV-2) immuno-affinity peptide enrichment LC-MS method (**A**). The LC-multiple reaction monitoring (MRM)/MS performance is based on RT-PCR results obtained from 48 positive and 308 negative samples. (**B**) The LC-MRM/MS performance based on all positive samples with an RT-PCR results below cycle threshold (Ct) 30 (limit of detection [LOD] for the LC-MRM/MS) and 308 negative samples.

(*Hoofnagle and Wener, 2009*) with LC-MRM/MS read-out. This is achieved by multiple factors that greatly outperform RT-PCR and rapid antigen test in theory. Firstly, antibodies are used to selectively enrich for the target peptide in a complex mixture. This helps increase the overall analytical sensitivity while LC-MS readily can distinguish between peptides in the separation and MRM steps. Secondly, internal standards added to the sample enable accurate and robust quantification. This provides an internal standard reference trace for every analyte and can help distinguish between false-positive chromatographic peaks based on retention time and ion ratios, that in RT-PCR experiment would be reported as a positive due to the absence of internal standards and since each gene is detected by a single reporter dye.

The sensitivity can be further improved by increasing the sample load if needed. Additionally, the number of viral protein targets can also be scaled by introducing additional anti-peptide antibodies into the sample mixture. This would allow for an LC-MS-based viral protein panel analysis method where relevant peptides, also including relevant spike peptides for mutation surveillance, are monitored in an endemic scenario, either covering new emerging SARS-CoV-2 strains or other viruses, such as influenza or respiratory syncytial viruses.

We show that the SISCAPA technology is an attractive route forward for future molecular pandemic surveillance systems. The accuracy of the LC-MS-based method would tolerate low levels of positive samples without compromising the positive predictive value of large-scale screening efforts, and thereby providing a next-generation platform for disease surveillance and an attractive alternative to today's RT-PCR-based technologies.

## Materials and methods
### Sample collection

The study was performed in accordance with the declaration of Helsinki and the study protocol (*Jämförande studier av Covid-19 smitta och antikroppssvar i olika grupper i samhället*) was approved by the Ethical Review Board of Linköping, Sweden (Regionala etikprövningsnämnden, Linköping, DNR – 2020–06395). Informed consent and consent to publish, including consent to publish anonymized data, was obtained from all subjects. Briefly, asymptomatic individuals working at an elderly caregiver in Sweden were screened on a regular basis at their workplace. A three-point collection (throat, nasal, saliva) was performed by participants using a self-sampling collection kit (Sansure Biotech, Changsha, China) containing PBS (1× PBS, 137 mM NaCl; 2.7 mM KCl; 4.3 mM $Na_2HPO_4$; 1.47 mM $KH_2PO_4$). Clinical samples were collected by swabs dipped into the sample collection tube and transported

to the laboratory within 8 hr. All samples were heat inactivated upon arrival to ensure that the core temperature of the vial reached at least 56°C for 30 min. The protocol used ensured that the core temperature did not reach above 60°C ± 0.5°C (1 sd), which has been shown to have no effect on the RT-PCR sensitivity.

### RT-PCR

Samples were analyzed using an RT-PCR test from Sansure Biotech (Changsha, China) according to FDA-EUA guidelines. The Novel Coronavirus (2019-nCoV) Nucleic Acid Diagnostic Kit was used for quantitative detection of the ORF-1ab and the N gene of novel coronavirus (2019-nCoV). Briefly, samples were lysed at room temperature for at least 10 min to allow for RNA release by chemical lysis using Sample Release Reagent (Sansure Biotech). The presence or absence of SARS-CoV-2 RNA was determined by RT-PCR combined with multiplexed fluorescent probing, which targets a SARS-CoV-2-specific region of ORF-1ab (FAM) and N gene (ROX) together with the human Rnase P internal control (Cy5). The RT-PCR analysis was performed using a CFX96 Real-Time PCR Detection System (Bio-Rad, Hercules, CA) programmed with the following RT-PCR protocol according to the manufacturer's instruction (50°C, 30 min; 95°C 1 min) followed by 45 cycles of (95°C 30 s, 60 °C 30 s). The RT-PCR results were interpreted according to instructions. Positive (FAM/ROX Amplification, Ct < 40). Negative (FAM/ROX No amplification; Cy5 Amplification, Ct < 40).

### Immuno-affinity peptide enrichment LC-MS

#### Materials

Recombinant NCAP was from R&D Systems, Minneapolis, MN, trypsin from Worthington, Lakewood, NJ, and anti-peptide antibodies from SISCAPA Assay Technologies, Washington, DC. All other chemicals were from MilliporeSigma, St Louis, MI, unless stated otherwise.

#### Calibrator preparation

NCAP digest, protocol described below, was used for calibration and quantitation of viral proteins. A serial dilution from 2.2 pmol/µl NCAP to 10,000, 2000, 400, 80, 16, and 3 amol/µl was performed consecutively in pooled negative sample background.

#### Samples

Clinical samples subjected to two freeze-thaw cycles prior were anonymized and two control pools were established by pooling randomly chosen samples based on their RT-PCR result (Ct < 30 [high pool], 30 ≤ Ct < 33 [low pool]). A total of 180 µl from each sample was used per enrichment experiment. A set of 48 positive and 308 negative samples was subjected to the LC-MS analysis.

### Protein extraction and digestion

A total of 20 µl of denaturant mixture (1 % (w/v) RapiGest [Waters Corporation, Milford, MA]) in 1 M triethylammonium bicarbonate, 50 mM dithiothreitol (Waters Corporation) were aliquoted into the collection plate (Waters Corporation). Next, 180 µl of the diluted NCAP and patient samples were carefully transferred from the collection tubes into the same plate. The plate was incubated on a heater-shaker at 500 rpm at 56°C for 15 min followed by the addition of 50 µl trypsin solution (7.3 mg/ml trypsin in 10 mM HCl). After mixing at 500 rpm for 30 s, the samples were digested at 37°C for 30 min and thereafter quenched by addition of trypsin stopping agent (0.22 mg/ml of Tosyl-L-lysyl-chloromethane hydrochloride in 10 mM HCl) at a final concentration of 37 µg/ml. The sample plate was mixed at 500 rpm for 30 s and incubated at room temperature for 5 min. The samples were spiked with 20 µl of SIL peptide mixture solution and mixed thoroughly on a shaker at 500 rpm for 30 s.

### Peptide enrichment

Anti-peptide antibodies, raised toward proteotypic peptides from the NCAP, were screened and validated as previously described (*Pope et al., 2009*). The antibody-coupled magnetic bead immune adsorbents corresponding to four SIL peptides (ADETQALPQR-$^{13}$C$_6$$^{15}$N$_4$, AYNVTQAFGR-$^{13}$C$_6$$^{15}$N$_4$, DGIIWVATEGALNTPK-$^{13}$C$_6$$^{15}$N$_2$, and NPANNAAIVLQLPQGTTLPK-$^{13}$C$_6$$^{15}$N$_2$) were resuspended fully by vortex mixing. The suspension of each anti-peptide antibody tube was mixed together in 1:1 ratio and 40 µl of the mixture was added to each digest. The plate was mixed at 1400 rpm to ensure that beads

were resuspended and thereafter incubated for 1 hr at 800 rpm at room temperature. After 1 hr incubation, the plate was placed on a magnet array (SISCAPA Assay Technologies). As soon as the beads had settled on the sides of each well (typically 1 min), the supernatant was removed; 150 µl of wash buffer (0.03% CHAPS, 1× PBS) was added to each sample and the beads were fully resuspending at 1400 rpm for 30 s and 450 rpm for another 30 s. The plate was placed on the magnet array again and the supernatant was removed. This step was repeated three times. The beads were subsequently resuspended in 50 µl elution buffer (0.5% formic acid, 0.03% CHAPS) and incubated for 5 min at room temperature. The beads were discarded by transferring the eluent to a QuanRecovery plate (Waters Corporation) for LC-MS analysis.

## LC-MS detection and quantification

Chromatography was performed on an ACQUITY UPLC I-Class FTN system, with Binary Solvent Manager and column heater (Waters Corporation); 20 µl of the enriched sample was injected onto a ACQUITY Premier Peptide BEH C18, 2.1 mm × 50 mm, 1.7 µm, 300 Å column (Waters Corporation) and separated using a gradient elution of mobile phase A containing laboratory LC-MS grade de-ionized water with 0.1% (v/v) formic acid, and mobile phase B containing LC-MS grade acetonitrile with 0.1% (v/v) formic acid. The gradient elution was performed at 0.6 ml/min with initial inlet conditions at 5% B, increasing to 28% B over 4.5 min, followed by a column wash at 90% B for 0.6 min and a return to initial conditions at 5% B. The total run time was 5.7 min, with a 6.5 min injection-to-injection cycle time.

A Xevo TQ-XS tandem MS (Waters Corporation, Wilmslow, UK) operating in positive electrospray ionization (ESI+) was used for the detection and quantification of the peptides. The instrument conditions were as follows: capillary voltage 0.5 kV, source temperature 150°C, desolvation temperature

**Table 2.** Multiple reaction monitoring (MRM) transitions and mass spectrometry (MS) method details target nucleocapsid protein (NCAP) severe acute respiratory syndrome coronavirus 2 (SARS-CoV-2) peptides.

| Peptide | MRM | MRM transition type | Cone voltage (V) | Collision energy (V) | Retention time (min) | Scan window (min) |
|---|---|---|---|---|---|---|
| ADETQALPQR | 564.8 > 400.2 | Quantifier | 35 | 19 | 1.09 | 0.6–1.4 |
| | 564.8 > 584.4 | Qualifier | 35 | 20 | | |
| | 564.8 > 712.4 | Qualifier | 35 | 24 | | |
| | 569.8 > 410.2 | SIL | 35 | 19 | | |
| AYNVTQAFGR | 563.8 > 679.4 | Quantifier | 35 | 19 | 2.49 | 2.0–3.0 |
| | 563.8 > 578.3 | Qualifier | 35 | 18 | | |
| | 563.8 > 892.5 | Qualifier | 35 | 19 | | |
| | 568.8 > 689.4 | SIL | 35 | 19 | | |
| DGIIWVATEGALNTPK | 562.3 > 643.4 | Quantifier | 35 | 14 | 4.12 | 3.6–4.8 |
| | 562.3 > 572.3 | Qualifier | 35 | 18 | | |
| | 562.3 > 700.4 | Qualifier | 35 | 14 | | |
| | 565.2 > 708.4 | SIL | 35 | 14 | | |
| NPANNAAIVLQLPQGTTLPK | 687.4 > 841.5 | Quantifier | 35 | 18 | 3.92 | 3.6–4.2 |
| | 687.4 > 766.4 | Qualifier | 35 | 23 | | |
| | 687.4 > 865.5 | Qualifier | 35 | 23 | | |
| | 690.4 > 849.5 | SIL | 35 | 18 | | |

600°C, cone gas flow 150 l/h, and desolvation gas flow 1000 l/h. The MS was calibrated at unit mass resolution for MS1 and MS2. Light and heavy labeled peptides were detected using MRM mode of acquisition with experimental details overviewed in *Table 2*.

TargetLynx XS (Waters Corporation) was used to process the raw LC-MS data, that is, signal processing (mean smoothing and background subtraction), peak detection (area and height), and quantification of the MRM chromatograms, including the calculation of the quantifier ion to qualifier ion ratio. The quantified data were exported as tables (*Supplementary file 1*) and additional analysis and visualization carried out using Python 3.

## Data and materials availability

The ProteomeXchange ID for this dataset is PXD026366. The proteomics data have been deposited to Panorama Public (*Sharma et al., 2014*) (https://panoramaweb.org/sars-cov-2_siscapa.url). This dataset includes raw files and integrated peak areas from TargetLynx XS, as well as visualization of all LC-MRM/MS chromatograms.

## Additional information

### Competing interests

Dominic Foley, Thomas McDonald, Johannes PC Vissers, Rebecca Pattison, Samantha Ferries, Sigurd Hermansson, Ingvar Betner, Amy Bartlett, Lisa Calton: employed by Waters Corporation. Morteza Razavi, Richard Yip, Matthew E Pope, Terry W Pearson, Leigh N Andersson: employed by SISCAPA Assay Technologies. The other authors declare that no competing interests exist.

### Funding

No external funding was received for this work.

### Author contributions

Andreas Hober, Data curation, Formal analysis, Investigation, Methodology, Supervision, Validation, Visualization, Writing - original draft, Writing - review and editing; Khue Hua Tran-Minh, Investigation, Methodology, Validation; Dominic Foley, Conceptualization, Data curation, Formal analysis, Investigation, Methodology, Validation; Thomas McDonald, Rebecca Pattison, Samantha Ferries, Conceptualization, Data curation, Formal analysis, Investigation, Methodology; Johannes PC Vissers, Data curation, Investigation, Methodology, Project administration, Software, Validation, Visualization, Writing - original draft, Writing - review and editing; Sigurd Hermansson, Data curation, Formal analysis; Ingvar Betner, Funding acquisition, Investigation, Supervision; Mathias Uhlén, Funding acquisition, Supervision, Writing - review and editing; Morteza Razavi, Richard Yip, Matthew E Pope, Terry W Pearson, Conceptualization, Methodology, Resources; Leigh N Andersson, Conceptualization, Methodology, Resources, Validation, Writing - review and editing; Amy Bartlett, Conceptualization, Formal analysis, Investigation, Methodology; Lisa Calton, Conceptualization, Data curation, Formal analysis, Investigation, Methodology, Project administration, Resources, Validation, Writing - original draft, Writing - review and editing; Jessica J Alm, Project administration, Resources, Writing - original draft, Writing - review and editing; Lars Engstrand, Investigation, Project administration, Resources, Writing - original draft, Writing - review and editing; Fredrik Edfors, Conceptualization, Data curation, Formal analysis, Investigation, Methodology, Project administration, Resources, Supervision, Validation, Visualization, Writing - original draft, Writing - review and editing

### Author ORCIDs

Andreas Hober http://orcid.org/0000-0001-8947-2562
Fredrik Edfors http://orcid.org/0000-0002-0017-7987

### Ethics

Human subjects: The study was performed in accordance with the declaration of Helsinki and the study protocol ("Jämförande studier av Covid-19 smitta och antikroppssvar i olika grupper i samhället") was approved by the Ethical Review Board of Linköping, Sweden (Regionala etikprövningsnämnden,

Linköping, DNR - 2020-06395). Informed consent and consent to publish, including consent to publish anonymized data, was obtained from all subjects.

### Decision letter and Author response
Decision letter https://doi.org/10.7554/eLife.70843.sa1
Author response https://doi.org/10.7554/eLife.70843.sa2

## Additional files

### Supplementary files
- Transparent reporting form
- Supplementary file 1. Integrated peak areas.

### Data availability
The ProteomeXchange ID for this dataset is PXD026366. The proteomics data have been deposited to Panorama Public (https://panoramaweb.org/sars-cov-2_siscapa.url), allowing for access to raw files and integrated peak areas from TargetLynx XS, as well as visualization of all LC-MRM/MS chromatograms.

The following dataset was generated:

| Author(s) | Year | Dataset title | Dataset URL | Database and Identifier |
|-----------|------|---------------|-------------|-------------------------|
| Edfors F | 2021 | Rapid and sensitive detection of SARS-CoV-2 infection using quantitative peptide enrichment LC-MS analysis | http://proteomecentral.proteomexchange.org/cgi/GetDataset?ID=PXD026366 | ProteomeXchange, PXD026366 |

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
