## [Editor Report]

With the prospect of pandemic readiness, the data presented here shows that MS can and most probably will start to become an essential analytical contribution.

---

## [Decision Letter]

**Decision letter after peer review:**

Thank you for submitting your article "Rapid and Sensitive Detection of SARS-CoV-2 Infection Using Quantitative Peptide Enrichment LC-MS/MS Analysis Running title: Quantitative Peptide Affinity LC-MS/MS Analysis of SARS-CoV-2" for consideration by *eLife*. Your article has been reviewed by 3 peer reviewers, including Maarten Dhaenens as Reviewing Editor and Reviewer #1, and the evaluation has been overseen by Y M Dennis Lo as the Senior Editor. The following individuals involved in review of your submission have agreed to reveal their identity: Jeroen Demmers (Reviewer #2); Marica Grossegesse (Reviewer #3).

Essential revisions:

As a guest editor, I have tried to organize the main concerns as described in more detail below. Please make sure to address these where possible. The original revisions provide a broader context of the requested reviews.

1. Sample preparation

a. How were the antibodies used in the present study made and characterized? Can you specify this in the material and methods section? (MG, JD) If proprietary and legal issues forbid the sharing of this data, we are willing to trust the overall performance.

b. It would be reasonable show with own data or to cite a publication stating that heat inactivation does not compromise the real-time PCR readout. (MG)

c. "50 μL elution buffer (0.5 % 180 formic acid, 0.03% CHAPS, 1X PBS) and incubated for 5 min at room temperature." This minor sentence is placed under major remarks, because in our understanding the elution buffer needs to be acidic and adding PBS will reduce acidity. If this is a typo, please correct. If this is not, could the authors try and use H2O instead and see if their results improve? (MD)

2. Figure 2: Peptide detection

a. What is represented in panels A and B. Is this the pure SIL peptide of the endogenous peptide in a complex matrix? What does '3 amol/ul' in the middle chromatogram exactly mean? (JD, MD)

b. Calibration curves:

i. I assume that the input is pure SIL peptide? (JD)

ii. Throughout the manuscript, could the calibration curves be displayed with a Log2-transformation in the X-axis to increase the resolution of the low abundant signal? Alternatively, or additionally provide a zoom inset to make it even more clear that 60 amol is still more signal than the negative sample. (MD)

c. Figure 2B: Add description of y-axis and place the text outside the line, so that the reader can see if there are any data points hidden. (MG)

d. "on-column amount of 60 amol." Because of the enrichment procedure, could the authors specify what initial conditions they spiked into the dilution series prior to enrichment. This would allow recalculation and avoid confusion about the correctness of the 60 amol on column claim (which in our hands is still detectable). (MD)

3. Patient population:

a. I oppose to the representation of the results and the claim of 86% sensitivity. (MD) Please see detailed revisions for an argumentation and for an alternative way of reproting your results. MG additionally pointed out during the Review evaluation session that true negatives can only be obtained from pre-Sars-Cov-2 era patients (prior sampling).

b. "asymptomatic individuals screened for ongoing disease": How were these obtained? (MD)

c. The majority was tested positive for SARS-CoV-2. This is very different from the percentages observed in regular testing facilities. How was the study group composed? Were these individuals who were already admitted to the hospital? (JD)

d. Authors conclude at the end of the Results section that patient samples were collected at an infectious stage (MG). "patient-collected samples" is a little misleading. Do you mean self-collected samples? (MG)

e. P5L98: why saliva samples? These actually are the hardest matrix for SISCAPA in our hands… If the triple sampling is what underlies the mentioning of saliva, I would prefer not to mention it separately, because its weight in the final assay cannot be assessed and several groups have already reported having difficulties with this matrix. (MD)

4. Bubble plot:

a. What is the size of the bubbles? (JD, MD)

b. Why was the highest signal reported in Figure 3 as a green dot at log2 MRM response of -6 was not called "positive" in the LCMS assay? (JD, MD) If manual peak inspection was used for diagnosis, please elaborate.

c. When counting dots, this only adds up to 55 sample, not 86 as depicted in the confusion matrix. (MD) Specify the number of samples (MG)

d. In Figure 3 the grey data points represent "not detected" or "inconclusively identified". What is meant by this? (MD, JD)

e. Calculate the correlation of the MS response to the cT value and add to graphic. (MG)

5. The access to the raw data was denied (MD, MG)

6. "For the LC-MS results, the lowest response divided by three was used". What is meant here? (MD, JD)

7. Please minimize downplaying qPCR in the text. (MG)

8. All reviewers would really appreciate extending the data beyond only one peptide. If impossible, please elaborate very clearly on the dangers and emphasize that this is only a proof of principle.

Overall, there was a general agreement amongst the reviewers that we are at a pivotal point in time for mass spectrometry in the clinic. Only with the highest standard data and the best possible representation of the results will we reach the common goal of showing the potential of MS.

*Reviewer #3 (Recommendations for the authors):*

The manuscript adds interesting and important knowledge to the field of virus-targeted proteomics. Still, in my opinion, some points need to be clarified or changed:

1) Line 72: Could you specify what rapid test detect nucleic acids? Rapid tests for my understanding are antigen tests detecting mainly the NCAP protein of SARS-CoV-2.

2) Line 72-78: The authors write about the sensitivity as a drawback of antigen tests. What about the specificity?

3) Line 103: The selectivity of antibodies is still a widely discussed topic. If the antibody is well characterized it can definitely contribute to a high specificity what leads me to my question: How were the antibodies used in the present study made and characterized? Can you specify this in the material and methods section?

4) Line 169/170: During sample preparation four different SARS-CoV-2 peptides were enriched. Were these also detected by MRM in the patient samples? Or to ask the other way around: why do you only show results for a single peptide?

5) Line 263: What do you mean with QC samples? Are these the two pools?

6) Line 279: Do you mean highest instead of lowest cT values here? Why do negatively diagnosed samples even a have a cT value (shouldn't they be negative)?

7) Figure 3A: Calculate the correlation of the MS response to the cT value and add to graphic.

8) Figure 3B: Specify the number of samples (N=?).

9) Line 302: What do you mean with "true sensitivity"?

Further general remarks:

10) The data has been uploaded to respective data repositories (ProteomeXchange and Panorama Public). I was able to access the Panorama Public data, but not the ProteomeXchange data. Could you please provide the login data?

11) Page 3, line 57: The expression "patient-collected samples" is a little misleading. Do you mean self-collected samples?

12) Line 77: Repetition that antigen tests are less sensitive (already stated in line 74).

13) Line 87: Typo: remove "="

14) Line 125f: In the study heat-inactivated samples were used. Hence, it would be reasonable show with own data or to cite a publication that heat inactivation does not compromise the real-time PCR readout.

15) Line 215: Do you mean "Results and Discussion" here? Otherwise I am missing the "discussion" section.

16) Line 220-222: remove blank line.

17) Figure 2B: Add description of y-axis and place the text outside the line, so that the reader can see if there are any data points hidden.

18) Table 2: typo in caption, replace "="by "-"

19) Line 326: Typo: outperform.

---

## [Author Response]

Essential revisions:As a guest editor, I have tried to organize the main concerns as described in more detail below. Please make sure to address these where possible. The original revisions provide a broader context of the requested reviews.1. Sample preparationa. How were the antibodies used in the present study made and characterized? Can you specify this in the material and methods section? (MG, JD) If proprietary and legal issues forbid the sharing of this data, we are willing to trust the overall performance.

We have included relevant references describing how SISCAPA polyclonal antibodies are raised and evaluated, but cannot reveal any proprietary information other than previously published work in this field. The antibody performance has been evaluated our hands, by assessing their LOD and LOQ by serial dilution of recombinant NCAP protein.

b. It would be reasonable show with own data or to cite a publication stating that heat inactivation does not compromise the real-time PCR readout. (MG)

We have added two references supporting this claim and we have also stated that the temperature core was validated in-house by repeatedly heating sample tubes in our ovens. We measured their outside and inside temperature as well as the ambient temperature of the oven over the course of 40 minutes.

c. "50 μL elution buffer (0.5 % 180 formic acid, 0.03% CHAPS, 1X PBS) and incubated for 5 min at room temperature." This minor sentence is placed under major remarks, because in our understanding the elution buffer needs to be acidic and adding PBS will reduce acidity. If this is a typo, please correct. If this is not, could the authors try and use H2O instead and see if their results improve? (MD)

We apologize for the typo. This should say H2O and has been updated in the new version of the manuscript.

2. Figure 2: Peptide detectiona. what is represented in panels A and B. Is this the pure SIL peptide of the endogenous peptide in a complex matrix? What does '3 amol/ul' in the middle chromatogram exactly mean? (JD, MD)

The calibration curve represents the AYN-peptide, which is serially diluted in matrix background with a constant spike-in of full-length NCAP-protein. We have clarified this in the figure legend.

b. Calibration curves:i. I assume that the input is pure SIL peptide? (JD)

The calibration curves represent digested NCAP protein spiked with SIL peptides. All samples are prepared in digestion buffer and thereafter enriched by the SISCAPA workflow prior LC-MS quantification. This has been clarified in the manuscript.

ii. Throughout the manuscript, could the calibration curves be displayed with a Log2-transformation in the X-axis to increase the resolution of the low abundant signal?

We have inserted a zoomed-in window resolving the lower concentration range in the curve.

Alternatively, or additionally provide a zoom inset to make it even more clear that 60 amol is still more signal than the negative sample. (MD)

This zoom has been provided in the updated version of the manuscript.

c. Figure 2B: Add description of y-axis and place the text outside the line, so that the reader can see if there are any data points hidden. (MG)

This has been done in the updated in the new version of the manuscript.

d. "on-column amount of 60 amol." Because of the enrichment procedure, could the authors specify what initial conditions they spiked into the dilution series prior to enrichment. This would allow recalculation and avoid confusion about the correctness of the 60 amol on column claim (which in our hands is still detectable). (MD)

We have removed the on-column amount from the results and discussion and we present the initial concentration as this is more relevant for clinical tests.

3. Patient population:a. I oppose to the representation of the results and the claim of 86% sensitivity. (MD) Please see detailed revisions for an argumentation and for an alternative way of reproting your results. MG additionally pointed out during the Review evaluation session that true negatives can only be obtained from pre-Sars-Cov-2 era patients (prior sampling).

We have followed the suggestion and will refer to this as Positive Percent Agreement (PPA). We have therefore changed sensitivity to PPA and NPA accordingly: Positive percent agreement (estimated sensitivity) and negative percent agreement (estimated specificity).

b. "asymptomatic individuals screened for ongoing disease": How were these obtained? (MD)

We have described the enrolment procedure in more detail to clarify this further.

c. The majority was tested positive for SARS-CoV-2. This is very different from the percentages observed in regular testing facilities. How was the study group composed? Were these individuals who were already admitted to the hospital? (JD)

A number of positive samples were specifically selected based on PCR-based results from a larger study. This has been specified under Samples (Immuno-Affinity Peptide Enrichment LC-MS).

d. Authors conclude at the end of the Results section that patient samples were collected at an infectious stage (MG). "patient-collected samples" is a little misleading. Do you mean self-collected samples? (MG)

We have changed the wording in the new version of the manuscript.

e. P5L98: why saliva samples? These actually are the hardest matrix for SISCAPA in our hands… If the triple sampling is what underlies the mentioning of saliva, I would prefer not to mention it separately, because its weight in the final assay cannot be assessed and several groups have already reported having difficulties with this matrix. (MD)

This is the standard procedure in Sweden. We have clarified that all samples were collected using the same triple point strategy, and that we were not mixing nasal swabs with saliva swabs. The mentioning of saliva by itself was not intended and has been removed in the updated version of the manuscript.

4. Bubble plot:a. What is the size of the bubbles? (JD, MD)

The size of the markers represents the (redundant) log2 MRM response of peptide AYNVTQAFGR. A fixed marker size has been used instead in the updated version.

b. Why was the highest signal reported in Figure 3 as a green dot at log2 MRM response of -6 was not called "positive" in the LCMS assay? (JD, MD) If manual peak inspection was used for diagnosis, please elaborate.

All green dots are missing the qualifier and thus do not pass the QC. This set of samples have been removed from the plot and are visualized in the boxplot instead. We have removed them from Figure 3A and are only visualizing them in Figure 3B.

c. When counting dots, this only adds up to 55 sample, not 86 as depicted in the confusion matrix. (MD) Specify the number of samples (MG)

Some of the markers overlay so attempting to count them may lead to a disconnect. The sample size has been added to the caption text and dots are only visualized in the boxplot (Figure 3B).

d. In Figure 3 the grey data points represent "not detected" or "inconclusively identified". What is meant by this? (MD, JD)

The grey dots represent PCR-positive samples where either the qualifier was missing (inconclusively) or nor the qualifier nor the quantifier were present (not detected). This has been added to the figure text.

e. Calculate the correlation of the MS response to the cT value and add to graphic. (MG)

The requested regression has been added to the figure.

5. The access to the raw data was denied (MD, MG)

We are really sorry for this misconception. The raw data is currently only available through Panorama with the provided login credentials as described above and in the Data Availability section. The PXD identifier is currently only reserved for the dataset. The data will though be available through ProteomeXchange after the review process.

6. "For the LC-MS results, the lowest response divided by three was used". What is meant here? (MD, JD)

We have excluded this visualization in the new version of the manuscript.

7. Please minimize downplaying qPCR in the text. (MG)

We have re-written the text to reflect a more neutral tone about the RT-PCR technology, which is truly amazing. We have removed the sentence listing all potential issues with the technology. The discussion is now centered around the sensitivity provided by the technology, which is subjected for criticism in relation to the clinical relevance (reference 11).

8. All reviewers would really appreciate extending the data beyond only one peptide. If impossible, please elaborate very clearly on the dangers and emphasize that this is only a proof of principle.

We have highlighted the limitations to this study in the conclusion-part of the paper. We have emphasized that this is a proof-of-principle study in the conclusion and discussed future applications.

Overall, there was a general agreement amongst the reviewers that we are at a pivotal point in time for mass spectrometry in the clinic. Only with the highest standard data and the best possible representation of the results will we reach the common goal of showing the potential of MS.Reviewer #3 (Recommendations for the authors):The manuscript adds interesting and important knowledge to the field of virus-targeted proteomics. Still, in my opinion, some points need to be clarified or changed:1) Line 72: Could you specify what rapid test detect nucleic acids? Rapid tests for my understanding are antigen tests detecting mainly the NCAP protein of SARS-CoV-2.

This sentence has been re-written to be more accurate. We have removed nucleic acids from the text.

2) Line 72-78: The authors write about the sensitivity as a drawback of antigen tests. What about the specificity?

This has been addressed in terms of binary read-out.

3) Line 103: The selectivity of antibodies is still a widely discussed topic. If the antibody is well characterized it can definitely contribute to a high specificity what leads me to my question: How were the antibodies used in the present study made and characterized? Can you specify this in the material and methods section?

We have added one sentence to the discussion where we explain that the selectivity of the assay is further improved by the mass spectrometer. We cannot present exactly how these antibodies have been generated, but have included references to the general SISCAPA workflow. The polyclonal antibodies have thus been validated in many different background matrices, and their sensitivity has been evaluated using recombinant NCAP protein serially diluted down to 3 amol/µl.

4) Line 169/170: During sample preparation four different SARS-CoV-2 peptides were enriched. Were these also detected by MRM in the patient samples? Or to ask the other way around: why do you only show results for a single peptide?

Only one antibody-peptide pair was selected for the cohort study. Only one peptide was selected, partly due to availability of reagents and multiplexing capability (amount of beads when eluting).

5) Line 263: What do you mean with QC samples? Are these the two pools?

Yes, this has been clarified in the manuscript. This was the peptide stored in elution buffer over the time course of two days (48 hours).

6) Line 279: Do you mean highest instead of lowest cT values here? Why do negatively diagnosed samples even a have a cT value (shouldn't they be negative)?

This section has been removed since the negative samples did not have any Ct value. This was imputed for visual purpose only.

7) Figure 3A: Calculate the correlation of the MS response to the cT value and add to graphic.

This has been added to the figure.

8) Figure 3B: Specify the number of samples (N=?).

This has been included in the new version of the manuscript in all figures.

9) Line 302: What do you mean with "true sensitivity"?

This has been updated in the new version, separating PPA, NPA from sensitivity, specificity.

Further general remarks:10) The data has been uploaded to respective data repositories (ProteomeXchange and Panorama Public). I was able to access the Panorama Public data, but not the ProteomeXchange data. Could you please provide the login data?

The ProteomeXchange data is hosted by Panorama and will be made available through the ProteomeXchange link after the review process. The current PXD-identifier is reserved for the dataset.

11) Page 3, line 57: The expression "patient-collected samples" is a little misleading. Do you mean self-collected samples?

This has been updated in the new version of the manuscript.

12) Line 77: Repetition that antigen tests are less sensitive (already stated in line 74).

This has been updated in the new version of the manuscript.

13) Line 87: Typo: remove "="

This has been removed.

14) Line 125f: In the study heat-inactivated samples were used. Hence, it would be reasonable show with own data or to cite a publication that heat inactivation does not compromise the real-time PCR readout.

This has been updated in the new version of the manuscript and we have provided references to relevant studies investigating this potential issue.

15) Line 215: Do you mean "Results and Discussion" here? Otherwise I am missing the "discussion" section.

This has been updated.

16) Line 220-222: remove blank line.

This has been removed.

17) Figure 2B: Add description of y-axis and place the text outside the line, so that the reader can see if there are any data points hidden.

This has been updated.

18) Table 2: typo in caption, replace "="by "-"

This has been updated.

19) Line 326: Typo: outperform.

This has been updated.